# Brief Communication: Widespread potential for seawater infiltration on Antarctic ice shelves

Sue Cook[1], Benjamin K. Galton-Fenzi[1,2], Stefan R. M. Ligtenberg[3], Richard Coleman[4,1]

[1]Antarctic Climate & Ecosystems Cooperative Research Centre, University of Tasmania, Private Bag 80, Hobart, Tasmania 7001
[2]Australian Antarctic Division, Channel Highway, Kingston, Tasmania 7050
[3]Institute for Marine and Atmospheric Research Utrecht (IMAU), P.O. Box 80005, 3508 TA Utrecht, The Netherlands
[4]Institute for Marine and Antarctic Studies, University of Tasmania, Private Bag 129, Hobart, Tasmania 7001, Australia

*Correspondence to*: S. Cook, sue.cook@utas.edu.au

**Abstract.** Antarctica's future contribution to sea level change depends on the fate of its fringing ice shelves. One factor which may affect the rate of iceberg calving from ice shelves is the presence of liquid water, including the percolation of seawater into permeable firn layers. Here, we present evidence that most ice shelves around Antarctica have regions where permeable firn exists below sea level. We find that seawater infiltration onto ice shelves may be much more widespread in Antarctica than previously recognised. Finally, we identify the locations where seawater infiltration is most likely to occur, with the intention that the results may be used to direct future radar studies.

## 1 Introduction

The infiltration of seawater into firn layers on an ice shelf has been observed in a number of locations around Antarctica. The process occurs as water from the ocean intrudes into pores between ice crystals in the firn layer (Scambos et al., 2009). Once within the firn, fresh water freezing turns the seawater into a layer of brine. Observations of brine in ice shelf firn layers have been made directly in firn cores on McMurdo Ice Shelf (Heine, 1968; Kovacs et al., 1982; Risk and Hochstein, 1967), Lazarev Ice Shelf (Dubrovin, 1960) and Brunt Ice Shelf (Thomas, 1975). Brine layers can also be observed in ice penetrating radar data. The brine causes strong absorption of the radar signal, and so can be identified by a reflective layer close to sea level (Smith and Evans, 1972) and a loss of basal return (Holland et al., 2009). Brine has been observed by this method on McMurdo (Grima et al., 2016), Wordie (Swithinbank, 1968), Wilkins (Vaughan et al., 1993), Larsen (Smith and Evans, 1972), Brunt (Walford, 1964) and Ross Ice Shelves (Neal, 1979; Robin et al., 1970).

The presence of liquid brine on an ice shelf changes its column-averaged density, thereby affecting calculations of ice thickness derived from altimetry. This in turn, will affect any inferred thickness of marine ice, or calculations of mass flux from the continent. The presence of liquid brine on an ice shelf has also been connected to increased fracture and calving (Scambos et al., 2009). Brine propagating laterally through the firn layer on an ice shelf may enter crevasses, where it can contribute to

crevasse penetration by hydrofracturing. Thus brine-enhanced fracturing could be an important consideration for predicting rates of iceberg calving from affected ice shelves.

Seawater infiltration can occur anywhere where permeable firn exists below sea level. Observations of brine on ice shelves have so far been sparse and opportunistic, with no comprehensive overview of where seawater infiltration may be an important consideration. In this paper we use pan-Antarctic geometry datasets along with results from the firn densification model IMAU-FDM (Ligtenberg et al., 2011) to map where the base of the permeable firn layer lies below sea level, referred to as the potential "brine zone". This represents the upper limit of where brine may be found on an ice shelf. We assess the reliability of the predicted area by comparison with reported observations, and discuss where future studies might be most likely to encounter brine layers on other ice shelves.

## 2 Methods

We use results from the IMAU-FDM firn densification model to estimate the depth of the permeable firn layer around the continent. The firn densification model IMAU-FDM simulates the temporal evolution of the firn density and temperature profile, including firn compaction rates and liquid water processes (percolation, retention, refreezing, and runoff of surface meltwater). The model is forced at the surface with climate data (e.g. surface temperature, precipitation, sublimation, and melt) from the RACMO2.3 regional climate model (Van Wessem et al., 2014).

Previous work has found that the permeability of firn "declines rapidly after the density of the firn reaches 750 kg m$^{-3}$" (Kovacs and Gow, 1975), while published values for the absolute pore close-off density range between 800 kg m$^{-3}$ (Kovacs and Gow, 1975) and 830 kg m$^{-3}$ (Cuffey and Paterson, 2010). Lower firn densities are more likely to support seawater infiltration, as they are less likely to have their pore spaces blocked by refreezing and their porosity means they can support a higher infiltration speed.

An average firn density profile over 1979-2013 is calculated to identify the depth of each of the three density layers below the surface. We then subtract this depth from the surface elevation to determine where permeable firn exists below sea level. We use surface elevations from five different datasets: Bedmap2 (Fretwell et al., 2013), CryoSat-2 (Helm et al., 2014), GLAS/ICESat (DiMarzio, 2007), RAMP (Liu et al., 2015) and REMA (Howat et al., 2018), allowing us to estimate uncertainties in the results. Using each elevation dataset we map out a potential "brine zone" for each of the three cut off densities, i.e. the area where permeable firn is predicted to exist below sea level. This brine zone is the theoretical areal extent where seawater infiltration is possible.

## 3 Results

Our results show that nearly all Antarctic ice shelves have a region where seawater infiltration is possible (Fig. 1, Table 1). The mean percentage of the total ice shelf area potentially vulnerable to seawater infiltration is 10% for the 750 kg m$^{-3}$ threshold, 22% for the 800 kg m$^{-3}$ threshold and 40% for the 830 kg m$^{-3}$ threshold, using ice shelf areas defined by the MOA

dataset (Haran et al., 2005; Scambos et al., 2007). These mean values are calculated from the results for each of the five surface elevation datasets, and have a standard deviation of 1.0 to 1.5 % of the total shelf area (a full table of results is available in the Supplementary material). The calculated brine extent is also susceptible to uncertainties in the FDM data, deriving from uncertainties within the FDM equations and long-term average accumulation and melt in RACMO2.3, which are typically around 15% (Ligtenberg et al., 2011). Using a combined firn depth uncertainty of 15%, and the quoted systematic bias of ±5 m in the Bedmap2 surface elevation dataset (Griggs and Bamber, 2011), we have also calculated upper and lower limits on the brine zone areas (Table 1).

## 4 Previous observations

To assess how the calculated brine zones compare to observations of true brine extent, we map out the locations of previous firn cores and radar surveys which can help to validate the presence of liquid brine (Fig. 2). The map shows all recorded firn cores on floating ice which have penetrated below sea level, most of which contained no brine. The exceptions are a number of ice cores on McMurdo Ice Shelf (Heine, 1968; Kovacs et al., 1982; Risk and Hochstein, 1967), one on Lazarev Ice Shelf (Dubrovin, 1960) and one on Brunt Ice Shelf (Thomas, 1975). All of the boreholes where brine has been observed lie within our predicted brine zones. The remaining ice cores, which penetrated below sea level, but contained no brine, are listed in Supplementary Table 4. We use these to provide a "false positive" rate for each of the predicted brine zones, referring to the proportion of these ice cores which our results predict should have contained brine. The 750 kg m$^{-3}$ zone has a false positive rate of 8%, the 800 kg m$^{-3}$ zone has a false positive rate of 42%, and the 830 kg m$^{-3}$ zone has a false positive rate of 67%.

We have only considered studies where an ice core was retrieved, as boreholes drilled with hot water may not reliably identify the presence of brine. We have also excluded one study on Shackleton Ice Shelf which stated that seawater was found in the "lower portion" (Morev et al., 1988). We believe this may refer to a brine-soaked layer, as the study found the firn layer to extend significantly below sea level, however it may also refer to upwelling when the core reached the ocean cavity so we have not included this as definitive evidence of seawater infiltration into firn on Shackleton Ice Shelf.

Seawater infiltration has also previously been identified on a number of ice shelves using ice penetrating radar, characterised by a reflective layer close to sea level and a loss of basal return (Fig. 2, Supplementary Table 5). Not all of these observations are useful for constraining the brine zone extent due to the imprecise published locations. In other cases, the ice front has changed significantly since the observations were made. For example, Wordie Ice Shelf has almost entirely collapsed since the brine observations were made (Cook and Vaughan, 2010).

The most comprehensive surveys of ice shelf brine layers come from a dense network of flights over Wilkins Ice Shelf, where a brine layer is present in almost the entire ice shelf area (Vaughan et al., 1993, Fig. 2b), and McMurdo Ice Shelf, where radar observations of brine have been confirmed by multiple firn core observations (Grima et al., 2016, Fig. 2e). The geometry of Wilkins Ice Shelf has changed substantially since the radar observations were made, however the 750 kg m$^{-3}$ brine zone still provides a reasonable match to the observed brine extent (Fig. 2b). Conversely, on McMurdo Ice Shelf the 750 kg m$^{-3}$ brine

zone over-predicts the observed brine extent (Fig. 2e). This is likely caused by the horizontal resolution of the datasets used (1 km for surface elevation, 27 km for firn depth) which limits our ability to represent small areas of seawater infiltration, and local climate effects. In all of the observational records we have studied, only one observation of brine has been made outside the predicted 750 kg m$^{-3}$ brine zone: an airborne radar campaign identified brine in the Western Ross Ice Shelf (Neal, 1979),

in a region lying within the 800 kg m$^{-3}$ brine zone (Fig. 2c). These results lead us to conclude that the 750 kg m$^{-3}$ brine zone provides the closest match to the true brine extent, although with substantial uncertainties.

## 5 Discussion

Our results identify many new locations where seawater infiltration is possible but has not been previously observed. This implies that seawater infiltration may be much more widespread than has previously been realised, although the uncertainties

on the mapped areas are high. The size of the potential brine zone is controlled by the balance between the thickness of the ice shelf and the depth of the firn layer. Some regions with a large potential brine zone have an unusually thick firn layer caused by high snowfall, such as Edward VIII Bay (Supplementary Figs. 1&2). In other areas such as the Riiser-Larsen area and Princess Ragnhild Coast, the large brine zone is caused by unusually thin ice, driven by the lateral divergence of ice shelves which are relatively unconfined, with only shallow embayments (Supplementary Fig. 3). On the West Antarctic coastline, high

snowfall is combined with significant ocean melting, both drawing permeable firn below sea level (Supplementary Fig. 2).

Density perturbations caused by an internal brine layer have the potential to affect variables calculated from altimetry data assuming hydrostatic equilibrium, e.g. ice shelf thickness, commonly used in ice sheet mass flux studies, or detection of basal marine ice (e.g. Fricker et al., 2001). The presence of brine can change the density of the firn layer by up to 70% (Heine, 1968), introducing error into calculated mass fluxes. The presence of brine on an ice shelf can also potentially affect iceberg calving

rates, through the availability of liquid water. Brine saturation has been identified as a potential factor behind the 2008 break-up of Wilkins Ice Shelf (Scambos et al., 2009). As liquid water on an ice shelf enters surface crevasses, the additional pressure causes the crevasses to deepen by hydrofracture, eventually leading to calving. Although seawater infiltration is a less efficient mechanism for hydrofracture than surface melting, since the brine column can be no higher than the waterline, it has the potential to enhance calving rates wherever it occurs. An interesting feature of our results is that the modelled potential brine

zone on Larsen C Ice Shelf aligns well with a zone of rift formation, which is the source of the much publicised crack which recently led to one of the largest calving events on record (Jansen et al., 2015) (Supplementary Fig. 6). The presence of brine in this area of Larsen C has been independently verified by an airborne radar survey, which detected brine "in the vicinity of rifts in the ice shelf." (Smith, 1972). This area of Larsen C experiences high rates of longitudinal spreading, which is likely the cause of both the rifting and the presence of brine, but it is unclear how far brine in the firn layer might affect the growth of

the rifts.

Our results indicate the total ice shelf area where permeable firn lies below sea level, but this does not necessarily imply that the firn contains brine. For seawater to enter the firn requires a direct connection between permeable firn and the ocean. This

means that for seawater infiltration to occur away from the calving front requires basal fractures in the ice to penetrate into the firn layer, connecting inland firn with ocean water. We hypothesise that this is the seawater pathway in the Western Ross Ice Shelf, where a brine layer has been observed at the "transition from valley glacier to ice shelf" (Robin et al., 1970) (Fig. 2c).

There is also the question of whether refreezing of brine within the firn layer will block pore spaces, preventing further penetration inland. This is the suggested reason behind observations from the Little America V firn core on the Ross Ice Shelf, where permeable firn at -22°C was found below sea level with no brine layer present (Gow, 1968). In sea ice literature, brine percolation has been observed to stop at a threshold of between -8 and -10°C (Golden et al., 2007; Pringle et al., 2009), but firn core data from McMurdo Ice Shelf has demonstrated that liquid brine can exist in firn at temperatures below -15°C (Kovacs et al., 1982; Risk and Hochstein, 1967). Therefore, low firn temperatures do not necessarily seem to be a barrier to seawater infiltration.

The extent of seawater infiltration is affected not only by the firn temperature, but also the speed of infiltration, which depends on the firn porosity, viscosity of the brine and the pressure gradient driving flow (Thomas, 1975). In order to maintain a steady state brine extent with seawater percolating inland from the coast, the speed of percolation should equal or exceed the speed of ice flow. Therefore, high flow speeds may prevent an ice shelf from maintaining a large area of liquid brine. However, if the seawater source is an inland fracture, the flow of the shelf is no barrier to a significant brine zone area. Previous observations of brine percolation speed are limited, but range from 245 ma$^{-1}$ on the McMurdo Ice Shelf (Kovacs et al., 1982) to 400 ma$^{-1}$ on the Brunt Ice Shelf (Thomas, 1975). Areas near an ice shelf calving front with high speeds and calving rates are unlikely to maintain a significant area of brine.

Taking these factors into account, we suggest the most likely new locations for brine to be observed are Abbot Ice Shelf, Nickerson Ice Shelf, Sultzberger Ice Shelf, Rennick Ice Shelf, and slower moving regions of Shackleton Ice Shelf. These shelves have large areas with permeable firn below sea level, relatively warm annual mean air temperatures and low flow speeds providing the ideal conditions for maintaining a large area of liquid brine. Many of these ice shelves have been overflown by Operation IceBridge or other airborne radar campaigns, and may be a potential target for identifying new brine regions. Other shelves such as the Drygalski Ice Tongue have an apparently large potential brine zone, but low firn temperatures are likely to limit seawater infiltration. In other locations such as Larsen D and West Ice Shelves, seawater infiltration is likely to occur but high ice velocities may limit the ability of the shelf to maintain a large brine zone.

**6 Conclusions**

Seawater infiltration has only previously been observed in a small number of ice shelves. Our results demonstrate that, according to firn densification model results, most ice shelves around Antarctica have some regions where permeable firn exists below sea level, meaning that seawater infiltration may be more widespread than previously realised. However, our mapped region of potential seawater infiltration is subject to high uncertainties arising from both the data used for mapping, and in the effects of other factors such as firn temperature on infiltration rates. We consider the most likely areas for future

observations of seawater infiltration to be Abbot Ice Shelf, Nickerson Ice Shelf, Sultzberger Ice Shelf, Rennick Ice Shelf, and slower moving regions of Shackleton Ice Shelf, and would recommend future work to examine airborne radar measurements over these regions for any evidence of a brine layer.

The area of ice shelves covered by a potential brine zone may also be set to grow in a warming climate. Melt rates underneath ice shelves have been observed to increase in recent decades, particularly in West Antarctica (Paolo et al., 2015). Increased basal melt would have the effect of drawing larger regions of permeable firn below sea level. In addition, increased snowfall (Thomas et al., 2015) would exacerbate this effect, although this is likely to be be counteracted by increasing surface melt in many locations (Kuipers Munneke et al., 2014). Some of the shelves we have identified in this study such as Abbot and George VI Ice Shelves and those on the Princess Ragnhild Coast are predicted to experience firn air depletion over the next two centuries, which would prevent further seawater infiltration (Kuipers Munneke et al., 2014). Widespread seawater infiltration could be an important consideration for future studies using altimetry to calculate Antarctic mass flux. It also has implications for the future stability of ice shelves, due to its potential to affect fracture propagation and contribute to ice shelf disintegration by hydrofracture.

## Acknowledgments

We thank J.L. Roberts and L.E. Peters for helpful discussion on interpreting geophysical data. DEMs were provided by NSIDC, the Byrd Polar and Climate Research Center and the Polar Geospatial Center under NSF-OPP awards 1543501, 1810976, 1542736, 1559691, 1043681, 1541332, 0753663, 1548562, 1238993 and NASA award NNX10AN61G. Computer time provided through a Blue Waters Innovation Initiative. DEMs produced using data from DigitalGlobe, Inc. This work was supported by the Australian Government's Business Cooperative Research Centres Programme through the Antarctic Climate and Ecosystems Cooperative Research Centre (ACE CRC) and by the Australian Research Council's Special Research Initiative for Antarctic Gateway Partnership (Project ID: SR140300001).

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

**Table 1: Percentage of total Antarctic ice shelf area covered by potential brine zone.** For each of the three threshold densities we provide the mean percentage area of Antarctic ice shelves where permeable firn lies below sea level, calculated using five different surface elevation models. Lower and upper bounds were calculated using maximum uncertainties in the firn air content (Ligtenberg et al., 2011)

10   and Bedmap2 surface elevation dataset (Griggs and Bamber, 2011).

| | Percentage of total ice shelf area | Standard deviation | Lower bound | Upper bound |
|---|---|---|---|---|
| Brine zone: 750 kg m$^{-3}$ | 10.2 % | 1.5 % | 1.3 % | 20.4 % |
| Brine zone: 800 kg m$^{-3}$ | 22.4 % | 1.2 % | 4.3 % | 52.8 % |
| Brine zone: 830 kg m$^{-3}$ | 40.5 % | 1.0 % | 8.7 % | 65.2 % |

15

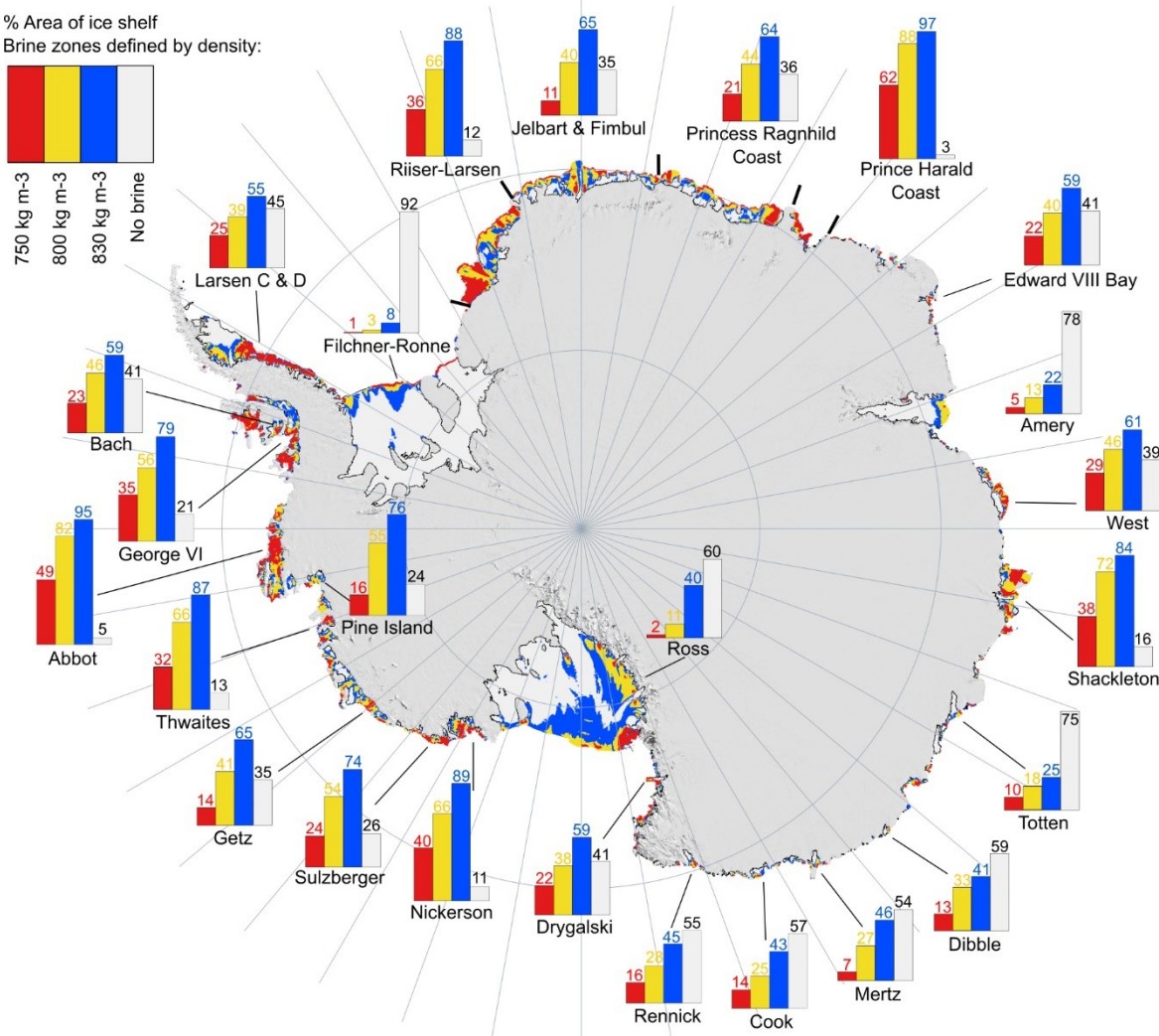

**Figure 1: Map of potential brine zones areas around Antarctica.** Map shows areas where permeable firn lies below sea level (the "brine zone"), with the threshold for firn permeability defined as 750 kg m$^{-3}$ (red), 800 kg m$^{-3}$ (yellow) and 830 kg m$^{-3}$ (blue) calculated using Bedmap2 surface elevation. Bar charts show the mean percentage area of selected ice shelves covered by the brine zone.

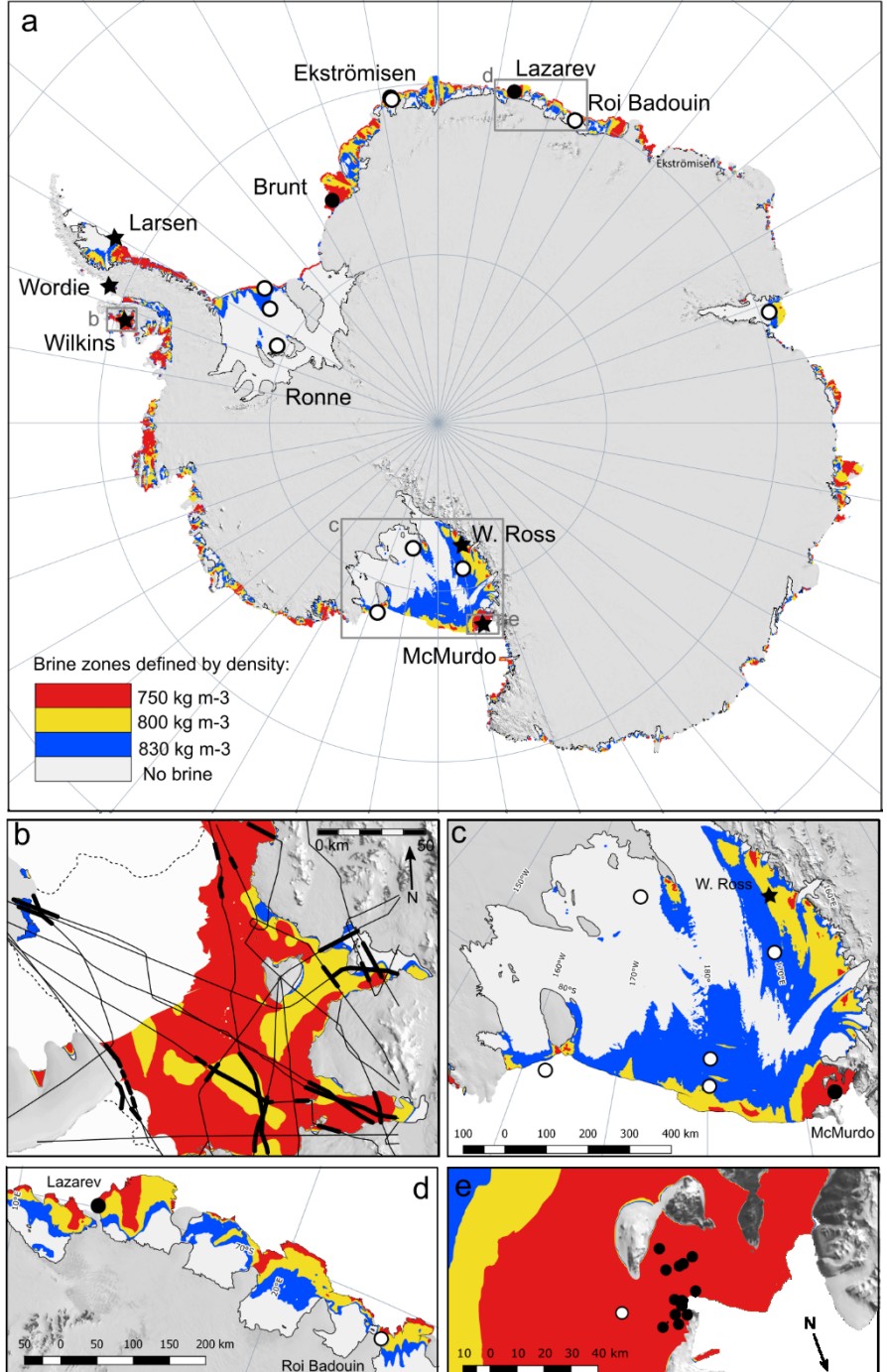

**Figure 2: Previous observations of firn and brine around Antarctica.** Coloured areas show potential brine zones as described in Fig. 1. Closed circles indicate boreholes where brine has been observed. Open circles are firn cores which penetrated below sea level but found no brine. Observations of brine using ice-penetrating radar are marked with black stars. Grey boxes on the main map indicate the extent of regions mapped in subfigures: **b**. Wilkins Ice Shelf, **c**. Ross Ice Shelf, **d**. Princess Ragnhild coast and **e.** McMurdo Ice Shelf. Subfigure **b** is adapted from (Vaughan et al., 1993) with thin black lines showing the full network of radar flight paths over the ice shelf. Thick black lines indicate areas where **no** brine was detected. Observations are described in detail in Supplementary Tables 4 & 5.