# Peer review of "Brief Communication: Widespread potential for seawater infiltration on Antarctic ice shelves"

_The Cryosphere, 2018_

## Referee Comment (RC1) · Anonymous Referee #1 · 20 Aug 2018

Review Summary: This manuscript presents an analysis of potential widespread nature of seawater infiltration into the firn of Antarctic ice shelves. The analysis utilizes the IMAU firn densification model forced by RACMO2.3 at the surface to assess the depth of three density horizons as being potentially permeable and thus enabling liquid water infiltration. When these horizons exist below sea level (assessed by differencing the FDM-determined density horizon depths with the Bedmap2 surface DEM), the authors characterize ice shelf areas as being potentially susceptible to ocean water infiltration. The authors appropriately acknowledge and quantify sources of uncertainty with the FDM, DEM, and density thresholds for permeability. Comparisons are made with available borehole and radar observations indicating and suggesting brine infiltration. Finally, the authors discuss the potential implications of brine infiltration and

mechanisms promoting or limiting the process. Although there are substantial (and well-acknowledged) uncertainties in the analysis, I believe this paper is well-crafted and identifies a process that is important, but typically overlooked in the analysis of Antarctic ice shelf stability. As such, I believe the paper should be published in The Cryosphere following relatively minor revisions.

My largest concern is with the comparisons with the validation datasets. These analyses are very qualitative in nature and I do not believe the authors suggest which density threshold for permeability is most likely to be correct. I suggest the authors more quantitatively present statistics related to the ground truthing (e.g., agreement, commission, and omission errors). Strengthening this side of the analysis would benefit the ice sheet modeling community and make the paper more impactful. Another concern is with the usage of the Bedmap2 DEM as the reason for its inclusion versus another DEM is not presented in the paper. Assessment of another DEM would be relatively straightforward and might offer insight into how much uncertainty is attributable to DEM choice. Finally, I offer some specific suggestions below on text and figure presentation.

Specific comments:

P2 Line 1 (and again near the end of the paper): The link to hydrofracturing is a bit unclear. Could you please describe in a sentence or two how bottom-up brine infiltration could contribute to (presumably) top-down hydrofracture?

P2 Line 28: Could you please add a citation for the 15% error associated with the FDM data?

P3 Line 11-12: This sentence is unclear, could you please restate it?

P3 Line 23+ and Figure 2: Areas of positive radar-derived brine identification are not clear. Are these the bold lines on Wilkins, whereas radar obs with no brine are thin lines? What about on McMurdo? What are the dashed lines? The labeling does not appear consistent between the subfigures. Please consider redrafting this figure to
make this clearer.

P5 Line 28: I don't believe this statement is fully justified. As the authors acknowledge, increased basal melting would draw down the surface of ice shelves, which will likely be experiencing increased surface melt (as indicated by the referenced Kuipers Munneke paper, as well as more recent studies). The Kuipers Munneke paper documents ice shelves likely to have exhausted their firn air content (FAC) by century's end. I would suggest the authors at least cross reference their results of brine infiltration with the areas that the Kuipers Munneke paper suggests will have their firn exhausted. Are the regions of potential brine infiltration also in areas where FAC will be exhausted? If so, this would suggest that brine infiltration is not likely to expand because ice shelves will be denser. The authors could add a sentence or two here with the results of this analysis, and it might provide information where brine infiltration will or will not be most likely in the future.

Figure 1: Could you also add error bars to the column plots?

[Figure]

---

## Referee Comment (RC2) · Anonymous Referee #2 · 4 Sep 2018

The manuscript uses firn densification modeling to assess where, in Antarctica, the permeable firn limit depth of Ice Shelves could lie below sea level (the so-called brine zone). The integrated possible brine zone area is estimated from 9% to 40% of the total area occupied by ice shelves in Antarctica. Ice Shelves with shear-margins have a buttressing effect slowing the flow of glaciers upstream. The disruption of ice shelves leads to faster delivery of grounded ice into the ocean, therefore increasing the rate of sea level rise. The possible lateral infiltration of sea water into permeable firn is a significant, so far under-considered, factor that could contribute to faster disruption of Ice Shelves.

The manuscript is an important contribution that should steer the research toward the brine infiltration mechanisms to constrain ice shelves stability. the manuscript is well

written and sound. I suggest publication in The Cryosphere after consideration of the following remarks.

**Firn layer for brine residency**

The authors study three firn layers where the brine could reside (750, 800, and 830 kg.mˆ{-3}). Considering these three firn layers is the largest source of uncertainty on the brine zone extent. To help qualitatively constrain this uncertainty, the authors should discuss where the brine is the most likely to reside. Using ice core observations, or radar-derived assessment for the brine depth found in the literature will help to feed such a discussion.

**Brine extent controlled by snow accumulation variability**

On the first paragraph p.5, the authors discuss the various processes that explain why the brine extent seems to overestimate the observed brine extents. The control of brine extent through snow accumulation is not clearly mentioned [Grima et al., 2019], while it does explain, at least, the western and eastern extension of the brine at SMIS. East of SMIS, downwrapping of internal layers has a significant role in controlling the brine extent. This setting will locally reduce the ice column over the brine layer, then reducing the horizontal pressure gradient to a point where it could eventually stop the brine propagation. On the West side, a negative surface mass balance is responsible for compact ice at the surface, limiting the brine extension.

**Regional Model Limits**

The SMIS brine described above is a good example showing how local singularities, that are not well accounted for by the continental data set used in the ms, might explained some mismatch between observed brine extent and the proposed brine zones. I suggest included such a remark in section 5.

**Minor Comments**

p.3-l.24-25. Please, cite the radio sounding papers providing brine layers extent at WIS

and MIS .

p.4-l.26. Could it also come from surface melting followed by downward migration?

Fig.2.a. The map contains more boxes (7) than actual sub-figures (4).

---

## Author Comment (AC1) · 30 Oct 2018

We thank both reviewers for their extremely helpful comments, which we have used to improve the text. Responses to the reviewers' comments are shown in blue below. The revisions to the manuscript have resulted in changes to line numbers and equivalent new line numbers are provided with each response.

**Anonymous Referee #1**

**Review Summary:** This manuscript presents an analysis of potential widespread nature of seawater infiltration into the firn of Antarctic ice shelves. The analysis utilizes the IMAU firn densification model forced by RACMO2.3 at the surface to assess the depth of three density horizons as being potentially permeable and thus enabling liquid water infiltration. When these horizons exist below sea level (assessed by differencing the FDM-determined density horizon depths with the Bedmap2 surface DEM), the authors characterize ice shelf areas as being potentially susceptible to ocean water infiltration. The authors appropriately acknowledge and quantify sources of uncertainty with the FDM, DEM, and density thresholds for permeability. Comparisons are made with available borehole and radar observations indicating and suggesting brine infiltration. Finally, the authors discuss the potential implications of brine infiltration and mechanisms promoting or limiting the process. Although there are substantial (and well-acknowledged) uncertainties in the analysis, I believe this paper is well-crafted and identifies a process that is important, but typically overlooked in the analysis of Antarctic ice shelf stability. As such, I believe the paper should be published in The Cryosphere following relatively minor revisions.

My largest concern is with the comparisons with the validation datasets. These analyses are very qualitative in nature and I do not believe the authors suggest which density threshold for permeability is most likely to be correct. I suggest the authors more quantitatively present statistics related to the ground truthing (e.g., agreement, commission, and omission errors). Strengthening this side of the analysis would benefit the ice sheet modeling community and make the paper more impactful. Another concern is with the usage of the Bedmap2 DEM as the reason for its inclusion versus another DEM is not presented in the paper. Assessment of another DEM would be relatively straightforward and might offer insight into how much uncertainty is attributable to DEM choice. Finally, I offer some specific suggestions below on text and figure presentation.

We have added ground truthing of the results, by including analysis of true/false positives. We have also expanded the text to discuss which is the most likely brine extent in the context of these results.

We have also added multiple alternative surface elevation models to improve our assessment of the uncertainty in the modelled brine extent. Instead of solely using the Bedmap2 surface elevation model to estimate brine extents, we have repeated the analysis using four alternative surface elevation models and adjusted the results to represent the mean of the five results (Fig.1, Table 1). We have also included the standard deviation of the different modelled extents (Table 1). The text has also been adjusted to reflect these changes (P2 lines 21-26, P3 lines 1-2).

**Specific comments:**
P2 Line 1 (and again near the end of the paper): The link to hydrofracturing is a bit unclear. Could you please describe in a sentence or two how bottom-up brine infiltration could contribute to (presumably) top-down hydrofracture?

Brine infiltration may occur both bottom-up, or laterally from the calving front. It is the second process which is linked to hydrofracture, as laterally propagating brine may enter any crevasses which penetrate below sea level. We have re-written this sentence to make clear which process we meant (P1 line 29).

P2 Line 28: Could you please add a citation for the 15% error associated with the FDM data?

Reference has been added (P3 Line 5).

P3 Line 11-12: This sentence is unclear, could you please restate it?

This line has been rewritten as part of the above changes to expand on the ground truthing of the calculated brine extents (P3 lines 14-17).

P3 Line 23+ and Figure 2: Areas of positive radar-derived brine identification are not clear. Are these the bold lines on Wilkins, whereas radar obs with no brine are thin lines? What about on McMurdo? What are the dashed lines? The labeling does not appear consistent between the subfigures. Please consider redrafting this figure to make this clearer.

The figure has been re-drawn to make the labelling of radar observations clearer. In subfigure a all radar observations are now marked with a star, and a description of the lines in subfigure b (Wilkins) has now been added to the caption.

P5 Line 28: I don't believe this statement is fully justified. As the authors acknowledge, increased basal melting would draw down the surface of ice shelves, which will likely be experiencing increased surface melt (as indicated by the referenced Kuipers Munneke paper, as well as more recent studies). The Kuipers Munneke paper documents ice shelves likely to have exhausted their firn air content (FAC) by century's end. I would suggest the authors at least cross reference their results of brine infiltration with the areas that the Kuipers Munneke paper suggests will have their firn exhausted. Are the regions of potential brine infiltration also in areas where FAC will be exhausted? If so, this would suggest that brine infiltration is not likely to expand because ice shelves will be denser. The authors could add a sentence or two here with the results of this analysis, and it might provide information where brine infiltration will or will not be most likely in the future.

This is an excellent point, and one which we did not discuss in sufficient detail in the original paper. As suggested, we have now included reference to ice shelves expected to experience firn air depletion (P6 l7-9). More detail on precisely how this would influence brine zone areas would require further study, particularly as there is a potential feedback as brine infiltration itself will increase rates of firn air depletion.

Figure 1: Could you also add error bars to the column plots?

We could not find a way to make error bars clearly legible to the reader. Instead we have provided the raw data in the Supplementary material (Supplementary Tables 1-3).

**Anonymous Referee #2**

The manuscript uses firn densification modeling to assess where, in Antarctica, the permeable firn limit depth of Ice Shelves could lie below sea level (the so-called brine zone). The integrated possible brine zone area is estimated from 9% to 40% of the total area occupied by ice shelves in Antarctica. Ice Shelves with shear-margins have a buttressing effect slowing the flow of glaciers upstream. The disruption of ice shelves leads to faster delivery of grounded ice into the ocean, therefore increasing the rate of sea level rise. The possible lateral infiltration of sea water into permeable firn is a significant, so far under-considered, factor that could contribute to faster disruption of Ice Shelves. The manuscript is an important contribution that should steer the research toward the brine infiltration mechanisms to constrain ice shelves stability. The manuscript is well written and sound. I suggest publication in The Cryosphere after consideration of the following remarks.

**Firn layer for brine residency**

The authors study three firn layers where the brine could reside (750, 800, and 830 kg.m^{-3}). Considering these three firn layers is the largest source of uncertainty on the brine zone extent. To help qualitatively constrain this uncertainty, the authors should discuss where the brine is the most likely to reside. Using ice core observations, or radar-derived assessment for the brine depth found in the literature will help to feed such a discussion.

We have included further discussion on how firn density affects brine infiltration in the methods section (P2 l18-20) to help readers understand which density is most likely to support substantial brine infiltration. We have also expanded the text to discuss more clearly which firn layer density the ice core data implies is most realistic (P3 l14-17, P4 l5-6). The value of 750 kg m$^{-3}$ is considered the most likely threshold from observations of brine infiltration rates on McMurdo ice shelf, as mentioned in the text (P2 line 16, Kovacs and Gow, (1975)) but as the brine substantially alters the density of the firn it infiltrates, observations made from ice cores cannot easily be used to narrow this further.

**Brine extent controlled by snow accumulation variability**

On the first paragraph p.5, the authors discuss the various processes that explain why the brine extent seems to overestimate the observed brine extents. The control of brine extent through snow accumulation is not clearly mentioned [Grima et al., 2019], while it does explain, at least, the western and eastern extension of the brine at SMIS. East of SMIS, downwrapping of internal layers has a significant role in controlling the brine extent. This setting will locally reduce the ice column over the brine layer, then reducing the horizontal pressure gradient to a point where it could eventually stop the brine propagation. On the West side, a negative surface mass balance is responsible for compact ice at the surface, limiting the brine extension.

The role of surface accumulation in controlling brine extent is mentioned at the beginning of the discussion section (P4, line 11-12) and is implicitly included in our calculations through the input of accumulation into the firn densification model. Our main limitation is in the resolution of the model results, which was not able to reproduce the local effects mentioned by the reviewer. To acknowledge this, we have added additional text discussing the impact of model resolution on our results (P4, lines 1-3).

**Regional Model Limits**

The SMIS brine described above is a good example showing how local singularities, that are not well accounted for by the continental data set used in the ms, might explained some mismatch between observed brine extent and the proposed brine zones. I suggest included such a remark in section 5.

This is an excellent point, and weave included a discussion of how the dataset resolution affects results in Section 4 (P4, lines 1-3).

**Minor Comments**

p.3-l.24-25. Please, cite the radio sounding papers providing brine layers extent at WIS and MIS.

References have been added (P3 l29-31).

p.4-l.26. Could it also come from surface melting followed by downward migration?

This airborne radar survey observed a loss of basal return, which they attributed in the paper to "penetration of brine into the firn layers through fractures at the borders of ice streams". A layer of refrozen surface water should not have caused loss of basal return in this way.

Fig.2.a. The map contains more boxes (7) than actual sub-figures (4).

The map has been redrawn for consistency in marking of features to remove this problem.

---

## Author Response (AR2)

Many thanks to the editor for these comments, which have helped us to improve the manuscript. We have described the changes made to the manuscript below (editor's comments in black, responses in blue).

Throughout the manuscript, the terms "seawater infiltration" and "brine infiltration" appear to be used interchangeably, e.g., lines 5/29 vs. 5/30. I find the former more evocative (and still accurate), but the latter is in more common use in the literature. If they are indeed referring to exactly the same phenomenon, and I believe they are, then only one such term should be used (I personally prefer "seawater").

This question has caused a lot of discussion amongst the authors as well! We have updated the text to use "seawater" to describe the initial infiltration process, but have retained the use of "brine" to describe the water existing within the firn, as it will have changed its salinity due to freshwater freezing after the initial infiltration process.

Abstract: The abstract could be improved in a couple of places. In particular, the term "variable" is not quite right, as it places the emphasis more on its use in a model than the "presence of liquid water", which is better characterized as a "physical property" of the ice column than a "variable". Separately, the last two sentences are written in the passive voice and could be more direct, e.g., "We find that seawater infiltration…" and "Finally, we identify the most likely locations…" Suggested changes have been made.

2/23: I understand the addition of additional DEMs as a response to reviewer #1, but some of them are now based on older data, e.g., Bamber et al. (2009) uses a combination of ERS-1 and ICESat data. Further, I was surprised to see that Helm et al. (2014, TC), which uses CryoSat-2 data, was not included, as it is publicly available and uses next-generation radar-altimetry data as compared to Bamber et al. (2009) or Liu et al. (2015). Further, since the Bedmap-2 surface DEM is – as best I understand it – intended to improve upon Bamber et al. (2009), that older DEM appears deprecated relative to Bedmap-2. Please reconsider which surface DEMs are most essential to include in this study. If the Bamber et al. (2009) DEM is kept, then it might be better to refer to it as ERS-1/ICESat to keep it more consistent with the other shorthand names.

We hadn't come across the surface DEM produced by Helm et al. (2014) before, but it is certainly a useful addition to the collection. We have replaced Bamber et al. (2009) with data from Helm et al. (2014), and updated the relevant tables in both the main manuscript and supplementary material. This has produced a small change in the mean brine zones, with a maximum absolute change of 0.4% total ice shelf area.

3/5: Aren't "error estimate" and "bias" simply uncertainties? In particular, a bias is not really a bias if its sign is unknown. If a single term can be used for all of the above, please do so.

We have slightly adjusted this sentence to improve clarity. The firn depth uncertainty is a simple uncertainty combined from all of the model inputs. The Bedmap2 uncertainties are provided as RMS random scatter in the data points (<15 m) and systematic bias (<5 m). Of these, only the systematic bias should have a significant effect on the calculated brine zones, but this bias will differ depending on location. Interpolation on ice shelves with sparse data points tends to produce a positive bias. The presence of marine ice, or higher than expected ice density will also produce a positive bias. However, divergence from hydrostatic equilibrium near the grounding line will give a negative bias. Rather than assume which effects dominate in which region, we applied the potential bias in each direction to encompass the two potential extremes.

4/13-14: Admittedly, the nomenclature gets tricky, but aren't "laterally unconfined ice shelves" simply "ice tongues"? I can't see from Figure S3 the names of these particular regions in detail, but if the local ice features at the mentioned coasts are already termed ice shelves, then this comment can be ignored.

This could have been phrased better. The ice shelves here aren't ice tongues, but they are in shallow embayments, so they have more lateral divergence than ice shelves which are more strongly confined (see attached figure). The sentence has been rewritten to make this more clear.

4/24: This is an interesting insight and useful context, but it's not made clear what the root cause of this model/observation alignment is. Based on the Supplementary Figures, in particular that for ice thickness, I suspect that an ice-thickness gradient there is the underlying cause for the different in predicted infiltration extent. Is that correct? Either way, further explanation would be appreciated.

We have added the following sentence: "This area of Larsen C experiences high rates of longitudinal spreading, which is likely the cause of both the rifting and the presence of brine, but it is unclear how far brine in the firn layer might affect the growth of the rifts."

5/22: IceBridge: This has been corrected

[Figure]

Scale shows flow speed,
from MEaSUREs v2
Arrows show flow direction

Quarisen
Quar Ice Shelf

Riiser-Larsen Ice Shelf

Riiser-Larsen Ice Shelf

0 m/yr
40 m/yr
80 m/yr
120 m/yr
160 m/yr
200 m/yr
240 m/yr
280 m/yr
320 m/yr
360 m/yr
400 m/yr
440 m/yr
480 m/yr
520 m/yr
560 m/yr
600 m/yr
640 m/yr
680 m/yr
720 m/yr
760 m/yr
800 m/yr

[Figure]

Princess Ragnhild Coast